# Decidua Basalis Mesenchymal Stem Cells Favor Inflammatory M1 Macrophage Differentiation In Vitro

**DOI:** 10.3390/cells8020173

**Published:** 2019-02-18

**Authors:** Mohamed H. Abumaree, Seham Al Harthy, Abdullah M. Al Subayyil, Manal A. Alshabibi, Fawaz M. Abomaray, Tanvier Khatlani, Bill Kalionis, Mohammed F. El- Muzaini, Mohammed A. Al Jumah, Dunia Jawdat, Abdullah O. Alawad, Ahmed S. AlAskar

**Affiliations:** 1Stem Cells and Regenerative Medicine Department, King Abdullah International Medical Research Center, King Abdulaziz Medical City, Ministry of National Guard Health Affairs, P.O. Box 22490, Riyadh 11426, Mail Code 1515, Saudi Arabia; ALSUBAYYILAB@NGHA.MED.SA (A.M.A.S.); khatlanita@NGHA.MED.SA (T.K.); jumahm@itkan-consulting.com (M.A.A.J.); jawdatd@ngha.med.sa (D.J.); askaras@ngha.med.sa (A.S.A.); 2College of Science and Health Professions, King Saud Bin Abdulaziz University for Health Sciences, King Abdulaziz Medical City, Ministry of National Guard Health Affairs, P.O. Box 3660, Riyadh 11481, Mail Code 3124, Saudi Arabia; 3National Center for Stem Cell Technology, Life Sciences and Environment Research Institute, King Abdulaziz City for Science and Technology, P.O Box 6086, Riyadh 11442, Saudi Arabia; ssalharthy@kacst.edu.sa (S.A.H.); Malshabibi@kacst.edu.sa (M.A.A.); alawad@kacst.edu.sa (A.O.A.); 4Department of Clinical Science, Intervention and Technology, Division of Obstetrics and Gynecology, Karolinska Institutet, 14186 Stockholm, Sweden; fawaz.abomaray@ki.se; 5Center for Hematology and Regenerative Medicine, Karolinska Institutet, 14186 Stockholm, Sweden; 6Department of Maternal-Fetal Medicine Pregnancy Research Centre and University of Melbourne. Department of Obstetrics and Gynaecology, Royal Women’s Hospital, Parkville, Victoria 3052, Australia; bill.kalionis@thewomens.org.au; 7Department of Obstetrics and Gynaecology, King Abdulaziz Medical City, Minstry of National Guard Health Affairs, P.O. Box 3660, Riyadh 11481, Mail Code 3124, Saudi Arabia; fmuzaini@hotmail.com; 8College of Medicine, King Saud Bin Abdulaziz University for Health Sciences, King Abdulaziz Medical City, Ministry of National Guard Health Affairs, P.O. Box 3660, Riyadh 11481, Mail Code 3124, Saudi Arabia; 9Adult Hematology and Stem Cell Transplantation, King Abdulaziz Medical City, Ministry of National Guard Health Affairs, P.O. Box 22490, Riyadh 11426, Mail Code 1515, Saudi Arabia

**Keywords:** placental stem cells, inflammation, M1 macrophages, inflammatory cells

## Abstract

Placental mesenchymal stem cells from maternal decidua basalis tissue (DBMSCs) are promising cells for tissue repair because of their multilineage differentiation and ability to protect endothelial cells from injury. Here, we examined DBMSC interaction with macrophages and whether this interaction could modulate the characteristics and functions of these macrophages. We induced monocytes to differentiate into M1-like macrophages in the presence of DBMSCs. DBMSC effects on differentiation were evaluated using microscopy, flow cytometry, and ELISA. DBMSC effects on M1-like macrophage induction of T cell function were also examined. The culture of DBMSCs with monocytes did not inhibit monocyte differentiation into M1-like inflammatory macrophages. This was confirmed by the morphological appearance of M1-like macrophages, increased expression of inflammatory molecules, and reduced expression of anti-inflammatory molecules. In addition, DBMSCs did not interfere with M1-like macrophage phagocytic activity; rather, they induced stimulatory effects of M1-like macrophages on CD4^+^ T cell proliferation and subsequent secretion of inflammatory molecules by T cells. We showed that DBMSCs enhanced the differentiation of M1-like inflammatory macrophages, which function as antitumor cells. Therefore, our findings suggest that DBMSCs are inflammatory cells that could be useful in cancer treatment via the enhancement of M1- like macrophages.

## 1. Introduction

Macrophages are leukocytes and function as antigen presenting cells with distinctive role in tissue homeostasis [1]. Macrophages are generally distributed throughout many tissues and help repair injured tissues in several human diseases [2]. Monocytes are recruited from the circulation into tissues by chemotactic signals initiated in response to physiological phenomena or pathological insults, where they differentiate into macrophages [2]. In addition, macrophages are also involved in the clearance of dead cells as part of their role in the later phases of tissue homeostasis and repair [2]. Macrophages are grouped into classically activated macrophages (M1 inflammatory macrophages) and alternatively activated macrophages (M2 anti-inflammatory macrophages) [2]. 

During acute inflammation, M1 macrophages are activated by agonists recognized by Toll-like receptors (TLRs) and by cytokines secreted from T helper 1 (Th1) cells, such as interferon-γ (IFN-γ). This increases their phagocytic activity against pathogens, stimulates their production of inflammatory cytokines, such as interleukin-1β (IL-1β), IL-12, and tumor necrosis factor-α (TNF-α); induces their production of reactive oxygen species; and enables their ability to present antigens to T cells through major histocompatibility complex (MHC) class II molecules [3,4]. In addition, M1 macrophages function as tumor suppressive cells through the activation of a Th1 cell inflammatory response against tumor cells [5]. 

Conversely, macrophages are induced to differentiate into the M2 phenotype by cytokines secreted by Th2 cells, such as IL-4 and IL-13. These cells are characterized by secretion of high levels of the anti-inflammatory cytokine IL-10, low levels of inflammatory cytokines, increased ability to inhibit the adaptive immune cell response, and increased cell-surface expression of CD14 (pattern recognition receptor), CD36 (class B scavenger receptor), CD163 (hemoglobin scavenger receptor), CD204 (class A scavenger receptor), CD206 (mannose receptor), and B7-H4 (co-inhibitory molecule) [2,3,4,6]. After an acute inflammatory phase, M2 macrophages become activated to resolve inflammation and induce tissue remodeling and wound repair [2,3,7]. In addition, M2 macrophages provide an anti-inflammatory microenvironment that is essential for tumor growth and progression [8]. 

Protective therapeutic effects of macrophages in response to various tissue injuries, including the central nervous system, have also been demonstrated [9]. For instance, M1 macrophages induce neuronal damage, whereas M2 macrophages promote neuronal regeneration [9]. In addition, since macrophages have an important role in the development of tumor, they are a target for potential therapeutic strategy [1]. Few potential strategies have been proposed to treat cancers by preventing the recruitment of macrophages into the microenvironment of tumor and promote the differentiation of macrophages into the anti-tumorigenic phenotype [1].

Mesenchymal stem cells, also known as multipotent stromal cells (MSCs), modulate the functions of innate and adaptive immune cells, including lymphocytes (T cells, B cells, and natural killer cells) and macrophages as well as dendritic cells [10]. We have previously shown that MSCs from chorionic villi of the human term placenta (pMSCs) induce an anti-inflammatory phenotype in human macrophages [6] and dendritic cells [11], and inhibit proliferation of T cells [11]. Furthermore, we previously reported the isolation and characterization of mesenchymal stem/multipotent stromal cells from maternal decidua basalis tissue (DBMSCs) of the human term placenta [12]. These DBMSCs express a unique combination of molecules involved in many important cellular functions including migration, proliferation, differentiation, immunomodulation, and blood vessel formation [12]. In addition, DBMSCs protect endothelial cells from injury induced by monocytes [13]. These phenotypic and functional characteristics make DBMSCs suitable for cellular therapy. However, the modulatory effects of DBMSCs on human macrophages are currently unknown. 

For DBMSCs to be used in cellular therapy, it is essential to determine their effects on the phenotypic properties and functional activities of macrophages. Here, we examined whether the interaction of DBMSCs with human monocytes would result in shifting monocyte differentiation from M1 inflammatory macrophages into M2 anti-inflammatory macrophages, and to determine the characteristic and functions of these differentiated macrophages. We found that DBMSCs enhanced the differentiation of M1 inflammatory macrophages, which produce inflammatory molecules that are known for their anticancer properties. In addition, DBMSCs induced M1 macrophage stimulation of CD4^+^ T cell proliferation and secretion of inflammatory cytokines. DBMSCs have the potential to treat cancer by inducing the anticancer activities of M1 macrophages directly on cancer cells or through T cell stimulation. 

## 2. Experimental Section

### 2.1. Ethics, Collection of Human Placentae, and Adult Peripheral Blood

The institutional review board at King Abdulla International Medical Research Centre (KAIMRC), Saudi Arabia, approved this study. Samples [placentae from uncomplicated human pregnancies (38–40 weeks of gestation) and peripheral blood samples from healthy adult participants] were used immediately after obtaining informed consent. KAIMRC research guidelines and regulations were followed to conduct all clinical and experimental procedures of this study.

### 2.2. Isolation and Culture of DBMSCs

MSCs were isolated from the decidua basalis (DBMSCs) of the maternal part of human term placenta using our previously published method [12]. DBMSCs were then cultured in a complete DBMSC culture medium (DMEM-F12 medium containing 10% MSCFBS (mesenchymal stem cell certified fetal bovine serum, catalogue number 12-662-011, Life Technologies, Grand Island, NE, USA), and antibiotics (100 µg/mL streptomycin and 100 U/mL Penicillin)), and then incubated at 37 °C in a humidified atmosphere containing 5% CO_2_ and 95% air (a cell culture incubator). DBMSCs (passage 3) of 30 placentae were used in this study.

### 2.3. Isolation of Human Monocytes

Mononuclear cells from peripheral blood (PBMNCs) of 30 normal healthy subjects were isolated using our previously published method [14]. Monocytes were isolated using human Monocyte Isolation Kit II (catalogue number 130-091-153, Miltenyi Biotec, Bergisch Gladbach, Germany) and a magnetic cell separation system (MACs: Miltenyi Biotec) as we previously described [14]. Trypan blue was used to determine the viability of monocytes while the purity was evaluated using anti-CD14 monoclonal antibody (R&D Systems, Systems, Abingdon, UK) in a flow cytometry. The purity was greater than 98%.

### 2.4. Culture of Monocyte Derived Macrophages with DBMSCs

M1 macrophages were differentiated from monocytes as we previously described [6]. Monocytes were seeded in 6-well plates in M1 macrophage differentiation medium (RPMI-1640 medium containing 50 ng/mL GM-CSF (R and D Systems), 10% FBS, 2 mM L-glutamine, and antibiotics indicated above), and then cultured at 37°C as described above for six days (Figure 1). For conditioned medium experiments (CMDBMSC), supernatant were prepared from the culture of unstimulated DBMSCs as previously described [6], and then added to the culture of monocytes (Figure 1). For the coculture experiments [soluble factor (SFDBMSC) and intercellular direct contact (ICDBMSC)], cells (DBMSCs and monocytes) were separated by transwell chamber membrane culture system (catalogue number 657640, ThinCert™ Cell Culture Inserts, Greiner Bio-One, Germany). For the experiments of SFDBMSC, DBMSCs were seeded on the upper compartments while monocytes were seeded in the lower compartment (Figure 1). For the experiments of ICDBMSC, DBMSCs were seeded on the reverse side of the membrane while monocytes were seeded on the upper side of the membrane (Figure 1). These two culture systems separate DBMSCs from monocyte-derived macrophages and therefore allow macrophage harvesting without contamination by DBMSCs. In all culture systems, cells were cultured in M1 macrophage differentiation medium (above). On Day 7, macrophages were harvested with TrypLE™ Express detachment solution (Life Technologies) and characterized using macrophage markers, MHC molecules, and costimulatory molecule (Table 1) in a flow cytometry technique as we previously described [6]. In some experiments, CMDBMSC, SFDBMSC, and ICDBMSC were added to the culture of monocyte-derived M1 macrophages on either Day 3 or Day 7 and then incubated for further three days and characterized as described above. In selected experiments, we added CMDBMSC, SFDBMSC, and ICDBMSC to the initial culture of monocyte- derived M1 macrophages incubated in RPMI-1640 medium without GM-CSF (spontaneous differentiation), incubated as described above, and the morphological changes were characterized by microscope examination. Different concentrations of DBMSC conditioned medium, 10, 20, 30, 40, 50, 60, 80, and 100% were tested. For SFDBMSC and ICDBMSC experiments, different ratios were used (1:1, 1:5, and 1:10 DBMSCs: monocyte-derived macrophages). Viability of macrophages were assessed using Trypan blue staining.

Before using DBMSCs in the coculture systems (SFDBMSC and ICDBMSC), DBMSCs were treated with 25 µg/mL Mitomycin C to inhibit their proliferation as we previously described [6]. To determine the reversibility of DBMSC effects on the differentiation of macrophages, CMDBMSC, SFDBMSC, and ICDBMSC were removed after three days of culturing with monocyte, and macrophages were then washed thoroughly and re-cultured in fresh M1 macrophage differentiation (above) for another three days. The negative control was monocyte-derived macrophages cultured alone in M1 macrophage differentiation. Experiments were carried out in duplicate and repeated 30 times using 30 individual preparations of both monocyte-derived macrophages and DBMSCs. 

### 2.5. Phagocytic Activity of Monocyte-Derived Macrophages

The phagocytic activity of monocyte-derived macrophages was studied using the CytoSelect™ phagocytosis kit (catalogue number CBA-224, Cell Biolabs, San Diego, CA, USA) as we previously published [6]. Briefly, macrophages were harvested from the three culture systems described above, and their phagocytic activities were then measured as we previously described [6]. Negative controls were monocyte-derived macrophages cultured alone in M1 macrophage differentiation, and macrophages cultured without Zymosan particles. Experiments were carried out in duplicate and repeated 10 times using 10 individual preparations of both monocyte-derived macrophages and DBMSCs. 

### 2.6. T Cell Proliferation Assay

Monocyte-derived macrophages cultured alone or cultured with DBMSCs in the culture system described above were harvested, washed with PBS, treated with 25 μg/mL Mitomycin C (described above), and different ratios of macrophages were then cultured with allogeneic CD4^+^ T cells (1:5, 1:10, and 1:20 macrophages: T cells) in triplicate in 96-well flat-bottomed plates. T cells were purified from PBMCs using CD4^+^ T Cell isolation Kit (catalogue number 130-096-533, Miltenyi Biotec) as we previously described [11]. All cultures were carried out in RPMI-1640 medium containing 10% FBS, 2 mM L-glutamine, and antibiotic as indicated above. After four days, T cell proliferation was measured using a tetrazolium compound [3-(4,5-dimethylthiazol-2-yl)-5-(3-carboxymethoxyphenyl)-2-(4-sulfophenyl)-2Htetrazolium, inner salt; MTS] kit (catalogue G5421, CellTiter 96® Aqueous Non-Radioactive Cell Proliferation Assay, Promega, Germany) as we previously described [11]. Experiments were carried out in triplicate and repeated 10 times using 10 individual preparations of T cells and macrophages harvested from 10 individual experiments of macrophages cultured with DBMSCs (CMDBMSC, SFDBMSC, and ICDBMSC). Results are presented as means of standard errors (± SE). T cell cultured alone served as a negative control.

### 2.7. Quantification of Human Cytokines

ELISA Kits (R&D Systems or MyBiosource, California, USA) were used according to the manufacturer’s instructions to quantify IL-1β, IL-6, IL10, IL-12, and IFN-γ in the supernatants obtained from macrophages cultured alone or with DBMSCs in the culture systems described above and from the T cell cultured alone or cultured with macrophages as described above. Complete RPMI-1640 and DMEM-F12 media were included as a negative control. 

### 2.8. Flow Cytometry

Cells (1 × 10^5^) were stained with antibodies listed in Table 1 for 30 min. Flow cytometry was then performed as we previously described [13]. Negative controls were cells stained with FITC or PE-labeled mouse IgG isotype antibody. 

### 2.9. Statistical Analysis

GraphPad Prism 5 was used to analyze data using non-parametric tests (Mann–Whitney U and Kruskal–Wallis). Data were deemed statistically significant if *p* < 0.05. 

## 3. Results and Discussion

### 3.1. DBMSCs Effect on M1-like Macrophage Differentiation from Human Monocytes

We used MSCs from decidua basalis of human term placenta (passage 3) as previously isolated and characterized by us [12]. DBMSCs at passage 3 are positive (> 95%) for MSC markers (CD44, CD90, CD105, CD146, CD166, HLA-ABC) and negative for hematopoietic markers (CD14, CD19, CD40, CD45, CD80, CD83, CD86, HLA-DR). DBMSCs at passage 3 also differentiate into adipocytes, chondrocytes and osteocytes [12]. Therefore, DBMSCs at passage 3 were used in all experiments. Monocytes were isolated from healthy human peripheral blood and induced to differentiate into M1-like macrophages using GM-CSF. After six days, cells exhibited a fried egg morphology a characteristic of M1-like macrophages (Figure 1A) [6]. These M1-like macrophages expressed CD14 (monocytic marker), but lacked expression of CD1a (dendritic cell marker) (data not shown). 

To study the effect of DBMSCs on macrophages, monocytes were cultured in an M1 macrophage differentiation medium in SFDBMSC and ICDBMSC culture systems at different cell ratios of macrophages: DBMSC (1:1, 10:1, and 20:1) and with 10, 20, 30, 40, 50, 60, 80, and 100% (*v*/*v*) CMDBMSCs. DBMSCs had no effect on the viability of macrophages as their viability was > 90%. Compared to untreated monocyte-derived macrophages, none of the DBMSC treatments inhibited the differentiation of monocytes into M1 macrophages, as cells exhibited a fried egg-shaped morphology indicative of M1 macrophages (Figure 1B–F). However, with increasing numbers and concentrations of DBMSCs, cells were smaller in size, showing a round morphology with reduced ability to adhere, suggesting that they were monocytes (Figure 1G–L). Cells that were differentiated in the presence of DBMSCs expressed CD14 in the absence of CD1a (data not shown). 

Next, we evaluated whether monocytes differentiated in the presence of DBMSCs at a ratio of 20:1 macrophages: DBMSC for both ICDBMSC and SFDBMSC culture systems and with 20% CMDBMSC expressed molecules characteristic of macrophages, as listed in Table 1. These experimental conditions (20:1 macrophages: DBMSC ratio and 20% CMDBMSC) were used in all subsequent experiments. Expression of these functional markers was studied using flow cytometry. Their expression was recorded as mean fluorescence intensity (MFI). After six days in culture, compared to untreated macrophages, DBMSCs (CMDBMSC or SFDBMSC) significantly increased the expression of CD14 and CD163 on macrophages, *p* < 0.05 (Figure 2A and B). In addition, DBMSCs (SFDBMSC) significantly increased the expression of CD206 on macrophages compared with that on untreated macrophages, *p* < 0.05 (Figure 2D). By contrast, ICDBMSCs significantly decreased expression of CD163, CD204, CD206, and CD36 on macrophages compared to untreated macrophages, *p* < 0.05 (Figure 2B–E), but there was no significant effect on the expression of CD14 and B7-H4, *p* > 0.05 (Figure 2A and F). Similarly, CMDBMSCs and SFDBMSCs had no significant effect on either the expression of CD204, CD36, or B7-H4 on macrophages compared with to untreated macrophages, *p* > 0.05 (Figure 2C,E, and F). Finally, CMDBMSCs did not significantly affect the expression of CD206 on macrophages compared to untreated macrophages, *p* > 0.05 (Figure 2D). 

Next, the effects of DBMSCs on macrophage differentiation were evaluated after adding DBMSCs to monocyte cultures on Day 3 or Day 7 and culturing for a further three days. All DBMSC treatments showed similar effects on M1-like macrophage differentiation of monocytes as described above (data not shown). These results suggest that DBMSCs affect macrophage differentiation at various times during culture. Similarly, all three DBMSC culture systems showed a similar effects on M1-like macrophage differentiation after coculture with monocytes in the absence of GM-CSF for seven days (data not shown), suggesting that DBMSCs possess immunostimulatory properties. 

### 3.2. DBMSC Effects on M1-like Macrophage Differentiation Are Irreversible

Next, we evaluated the reversibility of the effects of DBMSCs on the differentiation of macrophages. DBMSCs were removed from the monocyte cultures on Day 3, and monocyte-derived macrophages were then washed and cultured again in fresh M1-like macrophage differentiation medium without DBMSCs for a further three days. M1-like macrophage differentiation occurred under these experimental conditions, as evidenced by macrophages exhibiting the fried egg-shaped morphology of M1-like macrophages on Day 7. Moreover, these M1-like macrophages expressed similar levels of CD14, CD163, CD204, CD206, and B7-H4 to those of M1-like macrophages differentiated in the presence of DBMSCs for six days (Figure 3). These results suggest that the effect of DBMSCs on monocyte differentiation is irreversible.

### 3.3. DBMSC Effects on Expression of CD80, CD86, CD273, CD274, and HLA-DR on Macrophages

To determine the effect of DBMSCs on monocyte-derived macrophages with respect to macrophage function, a variety of immune functional markers (Table 1) were examined using flow cytometry, and expression was recorded as MFI. After six days in culture, DBMSC-treated macrophages in the three culture systems expressed significantly less CD86 than untreated macrophages, *p* < 0.05 (Figure 4C), Table 2. In addition, treatment with ICDBMSCs was associated with significantly decreased expression of co-inhibitory molecules (CD273 and CD274) on macrophages compared with that on untreated macrophages, *p* < 0.05 (Figure 4D,E), Table 2. HLA-DR (antigen presenting molecule) and CD80 (co-stimulatory molecule) expression did not differ between treated and untreated macrophages for any culture system, *p* > 0.05 (Figure 4A,B), Table 2. Similarly, CMDBMSC and SFDBMSCs did not significantly affect the expression of CD273 and CD274 on macrophages compared with that on untreated macrophages, *p* > 0.05 (Figure 4D,E), Table 2. These data indicate that ICDBMSC maintained the antigen presentation activity of macrophages and with immunostimulatory effect by maintaining the expression of co-stimulatory molecule (CD80) and reducing the expression of the co-inhibitory molecules (CD273 and CD274).

### 3.4. DBMSCs Modulate Expression by Macrophages of IL-1β, IL-6, IL-8, IL-12, IFN-γ, and TNF-α

Next, we determined the effects of DBMSCs on expression by macrophages of the six inflammatory cytokines listed in Table 1 using flow cytometry, with expression recorded as MFI. After six days in culture, treatment with ICDBMSCs resulted in significantly increased expression of IL-1β, IFN-γ, and TNF-α by macrophages relative to untreated controls, *p* < 0.05 (Figure 5A,E and F, Table 2. By contrast, ICDBMSCs significantly reduced the expression of IL-6 and IL-12 relative to controls, *p* < 0.05 (Figure 5B,D), while having no effects on the expression of IL-8, *p* > 0.05 (Figure 5C), Table 2. Similarly, compared to untreated macrophages, SFDBMSCs significantly reduced the expression by macrophages of IL-1β and IL-12, *p* < 0.05 (Figure 5A,D), while significantly increasing the expression of IFN-γ, *p* < 0.05 (Figure 5E), Table 2, and no effects on the expression of IL-6, IL-8, and TNF-α by macrophages were observed, *p* > 0.05 (Figure 5B,C, and F). In addition, CMDBMSCs significantly reduced the expression of IL-1β by macrophages relative to controls, *p* < 0.05 (Figure 5A), while having no effect on the expression of other inflammatory molecules, *p* > 0.05 (Figure 5B–F), Table 2.

### 3.5. DBMSC Effects on Expression by Macrophages of IDO, TGFβ1, TGFβ1, 2, 3, and HMOX-1

We additionally assessed the consequences of the interaction between DBMSCs and macrophages on the expression of the immunosuppressive enzyme indoleamine 2,3-dioxygenase (IDO) and the anti-inflammatory molecules transforming growth factor-β 1 (TGFβ1), TGFβ1, 2, 3, and heme oxygenase 1 (HMOX-1) using flow cytometry, with expression recorded as MFI or median percentage of positive cells for HMOX-1. After six days of culture with ICDBMSCs, expression of IDO by macrophages was significantly decreased compared with that by untreated macrophages, *p* < 0.05 (Figure 6A), Table 2. By contrast, CMDBMSCs and SFDBMSCs had no effect on expression of IDO by macrophages (Figure 6A), Table 2. In addition, none of the DBMSC treatments significantly modulated the expression of TGFβ1, TGFβ1, 2, 3, or HMOX-1 by macrophages compared with that by controls, *p* > 0.05 (Figure 6B–D), Table 2.

### 3.6. DBMSCs Modulate Secretion by Macrophages of IL-1β, IL-6, IL-10, and IL-12

We further characterized the effect of DBMSCs on secretion of cytokines by macrophages, including IL-1β, IL-6, IL-10, and IL-12 (p70), into the supernatant of the three culture systems described above using ELISA. Compared with untreated macrophages, treatment with DBMSCs from all three culture systems was associated with significantly increased secretion by macrophages of IL-6, *p* < 0.05 (Figure 7B), Table 3. In addition, CMDBMSCs significantly increased the secretion by macrophages of IL-10 relative to untreated controls, *p* < 0.05 (Figure 7C) Table 3, while ICDBMSCs significantly decreased the secretion of IL-10, *p* < 0.05 (Figure 7C) Table 3. None of the DBMSC treatments significantly affected secretion by macrophages of IL1β and IL-12, *p* > 0.05 (Figure 7A,D), Table 3. Likewise, secretion of IL-10 was not significantly altered by SFDBMSCs compared with that of untreated macrophages (Figure 7C), Table 3.

### 3.7. DBMSCs Do Not Alter Macrophage Phagocytic Activity

As described above, DBMSCs significantly modified the functions of macrophages by altering the expression of various immunological markers. Therefore, we tested whether DBMSCs affect the phagocytic activity of macrophages. For this purpose, phagocytic activity of macrophages cultured alone or initially cultured with CMDBMSCs, SFDBMSCs, or ICDBMSCs for six days was measured using the CytoSelect™ phagocytosis functional assay. None of the DBMSC treatments showed significant effects on the phagocytic activity of macrophages, *p* > 0.05 (Figure 8). 

### 3.8. DBMSCs Induce M1-like Macrophage Effects on T Cell Function

We further examined the combined effects of DBMSCs and M1-like macrophages on T cell function. Monocyte-derived M1-like macrophages that had either been cultured with DBMSCs (CMM1, SFM1, and ICM1) or cultured alone (M1) were added to allogenic CD4^+^ T cell cultures at T cell:M1-like macrophage ratios of 5:1, 10:1, and 20:1. T cell proliferation was then examined using the MTS assay. Compared to T cells alone, M1-like macrophages cultured either with DBMSC treatments (CMM1, SFM1, and ICM1) or without DBMSCs (M1) significantly increased T cell proliferation at all indicated ratios, *p* < 0.05 (Figure 9A). Similarly, ICM1 significantly increased T cell proliferation at all indicated ratios as compared to that with M1, *p* < 0.05 (Figure 9A). In addition, SFM1 significantly increased T cell proliferation at the 5:1 T cells: SFM1 ratio compared to that with M1, *p* < 0.05 (Figure 9A). By contrast, CMM1 significantly decreased T cell proliferation at the 10:1 and 20: 1 T cells: CMM1 ratios compared to that with M1, *p* < 0.05 (Figure 9A). Additionally, secretion of IL-10 by T cells stimulated with M1, CMM1, SFM1, and ICM1 was significantly reduced, *p* < 0.05 (Figure 9B), while the secretion of IL-12 and IFN-γ was significantly increased, *p* < 0.05 (Figure 9C,D), Table 4. 

### 3.9. Discussion

In this study, we examined the modulatory effects of DBMSCs on human macrophages, by coculturing DBMSCs with macrophages, which were generated from monocytes cultured in M1 macrophage differentiation medium [6]. Monocytes differentiated after six days into cells exhibiting the typical fried egg-shaped morphology of M1-like macrophages (Figure 1A) and expressed the monocytic marker CD14 without expressing the dendritic cell marker CD1a [6]. Addition of DBMSCs to monocyte cultures induced to differentiate into M1-like macrophages did not reverse M1 polarization nor shift it to M2 differentiation (elongated, spindle-like cells), as previously reported by us using pMSCs and by other researchers using MSCs from bone marrow, adipose tissues, gingival, or cord blood [6,15,16,17,18,19,20,21]. Morphologically, cells exhibited the typical morphology of M1-like macrophages (Figure 1B–F). We also observed that higher ratios of DBMSCs in the coculture inhibited monocyte differentiation into M1-like macrophages, as evidenced by the presence of cells smaller than macrophages with a round morphology and less adhesion, suggesting that these cells remained monocytes (Figure 1G–L). Phenotypically, M2-like macrophages express high levels of a number of cell surface markers, including CD14, CD36, CD163, CD204, CD206, and B7-H4, as we previously demonstrated with pMSCs [6]. In the presence of DBMSCs, M1-like macrophages upregulated the expression of only a subset (i.e. CD14, CD163, and CD206; Figure 2) of the markers. In this study, direct intercellular contact of DBMSCs (ICDBMSC) with monocyte-derived macrophages significantly decreased expression of CD36, CD163, CD204, and CD206 on macrophages, while paracrine communication between DBMSCs and monocyte-derived macrophages (SFDBMSC) increased the expression of CD14, CD163, and CD206 on macrophages (Figure 2B and D). In addition, soluble factors secreted by unstimulated DBMSCs (CMDBMSC) increased the expression of CD14 and CD206 on macrophages (Figure 2D). These data show the experimental conditions (mechanism of DBMSC communication with monocytes) modulate the phenotypic characteristics (inflammatory or anti-inflammatory phenotypes) of differentiated macrophages. However, at least with ICDBMSC, M1-like macrophages do not upregulate all markers, which suggests that ICDBMSCs are unable to induce an anti-inflammatory phenotype in macrophages, but rather they promote monocyte differentiation into inflammatory M1-like macrophages. One obvious explanation for this is that the original microenvironment (i.e., the niche) from which the MSCs were derived may determine the functional activities of MSCs on macrophage differentiation. DBMSCs are located in the maternal tissue of the placenta, which is continuously exposed to increased levels of inflammation and oxidative stress throughout human pregnancy [22,23]. By contrast, pMSCs are exposed to the fetal circulation, which experience relatively low levels of inflammation and exposure to oxidative stress mediators during normal pregnancy [24,25]. Likewise, bone marrow MSCs are generally exposed to low levels of oxidative stress in their bone marrow niche and only experience increased oxidative stress during injury or disease [26]. This phenomenon of monocyte polarization into immunostimulatory or immunoinhibitory macrophages was also reported for MSCs. It was shown that MSCs can be polarized into immunostimulatory or immunosuppressive cells similar to monocytes [27]. 

Kinetic experiments with monocyte-derived macrophages showed that the addition of all DBMSC treatments to monocytes cultured in M1 macrophage differentiation medium on Day 3 did not alter monocyte differentiation into M1-like macrophages, and was comparable to the effects of DBMSCs when added to the culture on the first day. Importantly, coculture with monocytes in M1 macrophage differentiation medium and DBMSCs from the time of plating up to Day 3 showed that the effects of DBMSCs on the differentiation of monocytes were not altered, as we observed the M1-like macrophage morphology was maintained at the end of the culture period. Hence, the effect of DBMSCs on the differentiation of M1-like macrophages is likely to be irreversible (Figure 3). 

CD163 is a hemoglobin scavenger receptor that mediates the resolution of inflammation by limiting damage induced by free hemoglobin [28] and stimulates the production of anti-inflammatory cytokines by M2-like macrophages [29,30]. Similarly, the scavenger receptor A (CD204), macrophage mannose receptor (CD206), and scavenger receptor B (CD36) are also involved in the resolution of inflammation by M2-like macrophages during the course of an innate immune response [31,32,33]. This further confirms that ICDBMSCs induced an inflammatory phenotype in macrophages. To support this observation, we demonstrated that ICDBMSCs decreased macrophage expression of CD273 and CD274 (Figure 4D,E), which are ligands for the inhibitory programmed death 1 (PD-1) receptor that signal to inhibit the function of T cells and to induce development of immunosuppressive regulatory T cells [34]. In addition, ICDBMSCs decreased macrophage expression of IDO (Figure 6A). IDO is an immunosuppressive enzyme that stimulates the catabolism of tryptophan, inhibiting the immune responses by repressing T cell function [35]. Moreover, ICDBMSCs decreased macrophage secretion of IL-10 (Figure 7C). IL-10 is an anti-inflammatory cytokine that inhibits the immune response by decreasing secretion of Th1 inflammatory cytokines and increasing the function of regulatory T cells [36]. Furthermore, ICDBMSC increased the secretion of IL-6 (Figure 7B). IL-6 activates the proliferative responses of lymphocytes, therefore shifting the immune response from an inhibitory to a stimulatory state [37]. These results suggest that intercellular contact between DBMSCs and monocyte-derived macrophages induces an inflammatory phenotype in macrophages. We further confirmed this by showing that ICDBMSCs increase the expression of inflammatory cytokines by M1-like macrophages, including IL-1β, IFN-γ, and TNF-α (Figure 5A,E and F). By contrast, ICDBMSCs decreased the expression of other inflammatory molecules by macrophages, including IL-6 and IL-12 (Figure 5B,D).

Additionally, we showed that paracrine communication between DBMSCs (SFDBMSC) and monocyte-derived macrophages increased the expression of IFN-γ (Figure 5E) and secretion of IL-6 (Figure 7B) while decreasing the expression of IL-1β and IL-12 (Figure 5A,D) by macrophages. Likewise, soluble mediators secreted by unstimulated DBMSCs (CMDBMSC) decreased the expression of IL-1β (Figure 5A) and increased the secretion of IL-6 and IL-10 by macrophages (Figure 7B,C). Collectively, our data indicate that DBMSCs may exert opposing effects on the phenotype of macrophages. 

Several other studies support our findings that soluble factors secreted by MSCs mediate immunomodulatory effects on immune cells. Others have reported that separation of MSCs and immune cells by a semipermeable membrane does not prevent the immunomodulatory effects that MSCs exert [6,38,39,40]. Importantly, our results also demonstrate that the immune/inflammatory effects of DBMSCs on macrophages are mediated by soluble factors secreted by unstimulated DBMSCs, suggesting that direct crosstalk between DBMSCs and immune cells is unnecessary, as we previously reported [6]. However, this is in contrast with previous studies reporting that it is essential for MSCs to be activated in order to modulate responses of immune cells [20,41,42]. 

Activated M1-like macrophages usually phagocytose microbes, remove tumor cells, present antigen to T cells to trigger an immune response, and secrete high levels of inflammatory cytokines [43]. Although all DBMSC treatments did not affect expression of HLA-DR (antigen-presenting molecule) and CD80 (costimulatory molecule), and decreased the expression of another costimulatory molecule, CD86, on M1-like macrophages, they did not inhibit the phagocytic activities of M1-like macrophages (Figure 8). In addition, all DBMSC treatments induced stimulatory effects of M1-like macrophages on the proliferation (Figure 9A) and production of high levels of inflammatory cytokines, including IL-12 and IFN-γ, and low levels of IL-10 (Figure 9B–D) by CD4^+^ T cells. These data demonstrate that although DBMSCs (ICDBMSC, SFDBMSC, and CMDBMSC) have differential effects on the inflammatory phenotype in M1-like macrophages, they induced similar modulatory effects on the functional activities of CD4^+^ T cells. This finding is important, because it suggests that DBMSCs are immune stimulatory cells with the potential for fighting cancer cells through their ability to induce the differentiation of M1-like macrophages, which have anticancer activity [5,44]. Additionally, DBMSCs induce an M1-like macrophage stimulatory effect on the function of CD4^+^ T cells, which also fight cancer cells [45]. The mechanisms by which DBMSCs stimulate M1 macrophage function are currently unknown. However, DBMSCs produce a large range of molecules that are involved in M1 macrophage differentiation. For example, DBMSCs produce IFN-γ and GM-CSF, which can induce M1 macrophage differentiation [46,47]. Therefore, these molecules may mediate the DBMSC effect on M1-like macrophages. However, further studies are required to elucidate these mechanisms and to determine the anticancer activities of DBMSCs. 

## 4. Conclusions

This is the first study to show that human DBMSCs favor differentiation of human M1 macrophages by direct or indirect contact mechanisms or through soluble factors secreted by unstimulated DBMSCs. We show for the first time that DBMSCs induce an inflammatory phenotype in M1 macrophages with functional activities that induce the T cell proliferation and cytokine secretion. In addition, we show that the effect of DBMSCs on macrophage differentiation is irreversible. However, a future study will be conducted to confirm this in vitro finding of DBMSCs favoring M1 macrophage differentiation by performing an in vivo study. In light of these results, we propose that human DBMSCs are immunostimulatory cells with the potential to target cancer cells by acting on macrophages to enhance macrophage anticancer activities either directly or indirectly through CD4^+^ T cells. 

## Figures and Tables

**Figure 1 cells-08-00173-f001:**
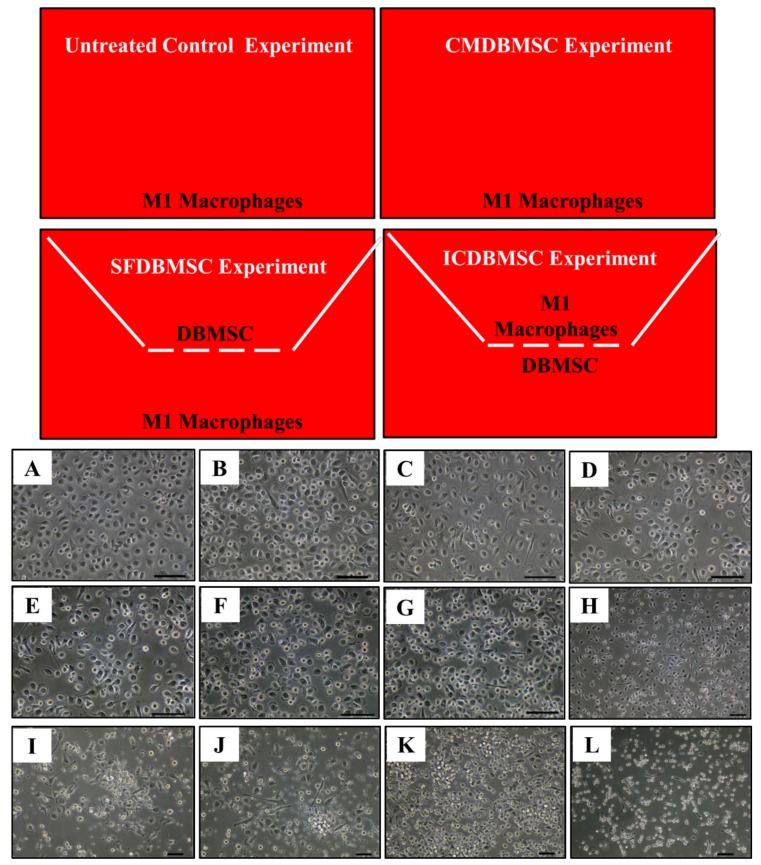
M1 macrophage culture system. Untreated control experiment consisted of M1 macrophages cultured on a surface of 6-well culture plate in a medium containing GM-CSF, CMDBMSC experiment consisted of M1 macrophages cultured on a surface of 6-well culture plate in a medium containing GM-CSF and CMDBMSC (conditioned medium), SFDBMSC (soluble factor), and ICDBMSC (intercellular direct contact) experiments. In SFDBMSC and ICDBMSC experiment, cells (DBMSCs and monocytes) were separated by transwell chamber membrane culture system. For the experiments of SFDBMSC, DBMSCs were seeded on the upper compartments while monocytes were seeded in the lower compartment. For the experiments of ICDBMSC, DBMSCs were seeded on the reverse side of the membrane while monocytes were seeded on the upper side of the membrane. GM-CSF medium was added to SFDBMSC and ICDBMSC experiments. Effects of human DBMSCs on the morphology of human monocytes differentiated into macrophages by GM-CSF. (**A**–**F**) Represent phase-contrast microscopic images showing monocyte (round-shaped morphology) differentiation into M1-like macrophages (fried egg-shaped morphology) after six days of culture in a medium containing GM-CSF (**A**), in a medium containing GM-CSF and DBMSCs at a 20:1 monocyte: DBMSC ratio (**B**), at a 10:1 monocyte: DBMSC ratio (**C**), at a 1:1 monocyte: DBMSC ratio (**D**), in the presence of 10% CMDBMSC (**E**), or in the presence of 20% CMDBMSC (**F**). (**G**–**K**) Representative phase-contrast microscopic images showing monocyte-like cells after six days of culture in a medium containing GM-CSF and 30% CMDBMSC (**G**), 40% CMDBMSC (**H**), 50% CMDBMSC (**I**), 60% CMDBMSC (**J**), 80% CMDBMSC (**K**), or 100% CMDBMSC (**L**). Experiments were carried out in duplicate and repeated 30 times using 30 individual preparations of both monocyte-derived macrophages and DBMSCs. Scale bars represent 50 µm.

**Figure 2 cells-08-00173-f002:**
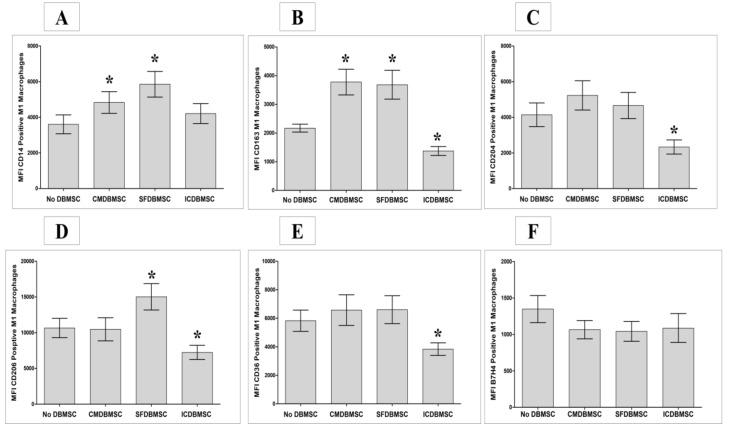
Effects of human DBMSCs on the expression of cell surface molecules CD14, CD163, CD204, CD206, CD36, and B7H4 on human monocytes differentiated into macrophages by GM-CSF, as analyzed by flow cytometry. After six days in culture, compared to untreated macrophages, CMDBMSCs significantly increased expression of CD14 (**A**) and CD163 (**B**) on macrophages while having no significant effect (*p* > 0.05) on expression of CD204 (**C**), CD206 (**D**), CD36 (**E**), and B7H4 (**F**) on macrophages. Compared to untreated macrophages, SFDBMSC significantly increased expression of CD14 (**A**), CD163 (**B**), and CD206 (**D**) on macrophages while having no significant effects (*p* > 0.05) on expression of CD204 (**C**), CD36 (**E**), and B7H4 (**F**) on macrophages. In addition, ICDBMSCs significantly decreased expression of CD163 (**B**), CD204 (**C**), CD206 (**D**) and CD36 (**E**) on macrophages while having no significant effects (*p* > 0.05) on expression of CD14 (**A**) and B7H7 (**F**) compared with that on untreated macrophages. Levels of expression are presented as median fluorescent intensity (MFI) as determined by flow cytometry. Experiments were carried out in duplicate and repeated 30 times using 30 individual preparations of both monocyte-derived macrophages and DBMSCs. * *p* < 0.05. Bars represent standard errors.

**Figure 3 cells-08-00173-f003:**
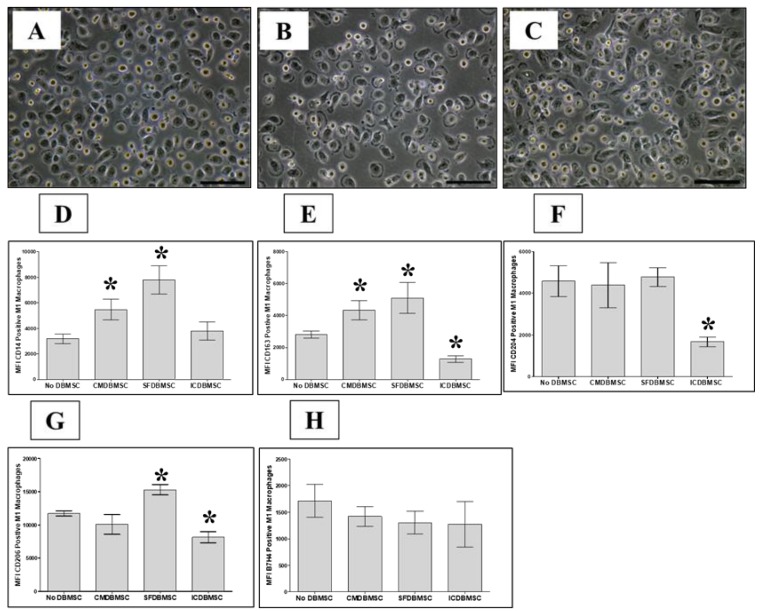
Reversibility effects of DBMSCs on macrophage differentiation. Human monocytes differentiated into macrophages by GM-CSF in the presence of DBMSCs. DBMSCs were removed from the monocyte cultures on Day 3, and monocyte-derived macrophages were then washed and cultured again in fresh M1-like macrophage differentiation medium without DBMSCs for a further three days. Macrophage differentiation was analyzed by morphological analysis using microscopic examination (Panels **A**, **B**, and **C**) and flow cytometric analysis of cell surface molecules (CD14, CD163, CD204, CD206, and B7-H4). (**A**–**C**) Representative phase-contrast microscopic images showing monocyte (round-shaped morphology) differentiation into M1-like macrophages (fried egg-shaped morphology) after six days of culture (**A**), in a medium containing GM-CSF and DBMSCs at a 20:1 monocyte: DBMSC ratio (**B**), in the presence of 20% CMDBMSC (**C**)**.** After six days in culture, compared to untreated macrophages, CMDBMSCs significantly increased the expression of CD14 (**D**) and CD163 (**E**) on macrophages, but had no significant effect on the expression of CD204 (**F**), CD206 (**G**), and B7-H4 (**H**) on macrophages. Compared to untreated macrophages, SFDBMSC significantly increased the expression of CD14 (**D**), CD163 (**E**), and CD206 (**G**) on macrophages, but had no significant effect on the expression of CD204 (**F**), and B7-H4 (**H**) on macrophages. In addition, ICDBMSCs significantly decreased the expression of CD163 (**E**), CD204 (**F**), CD206 (**G**) on macrophages, but had no significant effect on the expression of CD14 (**D**) and B7-H7 (**H**) compared with the effect on untreated macrophages. Levels of expression are presented as median fluorescent intensity (MFI) as determined by flow cytometry. Experiments were carried out in duplicate and repeated 10 times using 10 individual preparations of both monocyte-derived macrophages and DBMSCs. * *p* < 0.05. Bars represent standard errors.

**Figure 4 cells-08-00173-f004:**
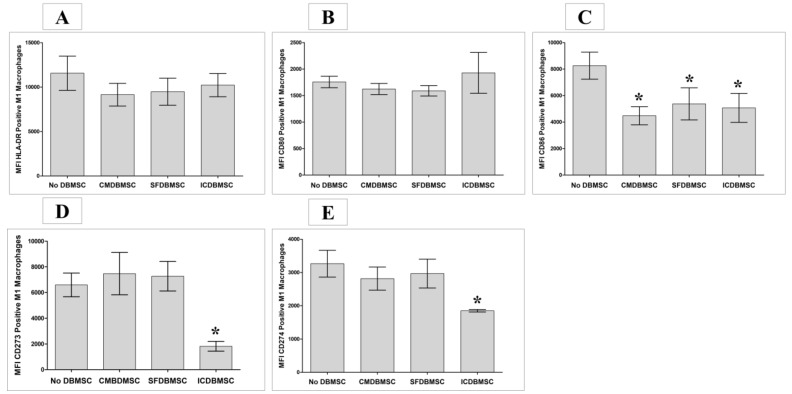
Effects of human DBMSCs on expression of the cell surface molecules HLADR, CD80, CD86, CD273, and CD274 by human monocytes differentiated into macrophages by GM-CSF, as analyzed by flow cytometry. After six days in culture, compared to untreated macrophages, CMDBMSCs and SFDBMSCs had no significant effects (*p* > 0.05) on expression of HLA-DR (**A**), CD80 (**B**), CD273 (**D**), and CD274 (**E**) on macrophages while significantly decreasing expression of CD86 (**C**). Compared to untreated macrophages, ICDBMSCs had no significant effects (*p* > 0.05) on expression of HLA-DR (**A**) and CD80 (**B**) on macrophages while significantly decreasing expression of CD86 (**C**), CD273 (**D**), and CD274 (**E**) on macrophages. Levels of expression are presented as median fluorescent intensity (MFI) as determined by flow cytometry. Experiments were carried out in duplicate and repeated 30 times using 30 individual preparations of both monocyte-derived macrophages and DBMSCs. * *p* < 0.05. Bars represent standard errors.

**Figure 5 cells-08-00173-f005:**
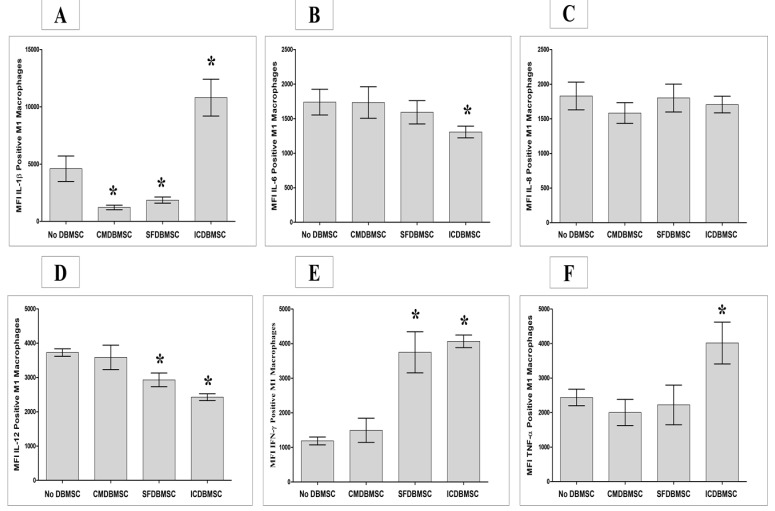
Effects of human DBMSCs on expression of the intracellular cytokines IL-1β, IL-6, IL-8, IL-12, IFN-γ, and TNF-α by human monocytes differentiated into macrophages by GM-CSF, as analyzed by flow cytometry. After six days in culture, compared to untreated macrophages, CMDBMSCs significantly decreased expression of IL-1β (**A**) by macrophages while having no significant effects (*p* > 0.05) on expression of IL-6 (**B**), IL-8 (**C**), IL-12 (**D**), IFN-γ (**E**), and TNF-α (**F**) by macrophages. Compared to untreated macrophages, SFDBMSCs significantly decreased expression of IL-1β (**A**) and IL-12 (**D**) while having no significant effects (*p* > 0.05) on expression of IL-6 (**B**), IL-8 (**C**), and TNF-α (**F**), but significantly increasing expression of IFN-γ (**E**) by macrophages. In addition, compared to untreated macrophages, ICDBMSCs significantly increased expression of IL-1β (**A**), IFN-γ (**E**), and TNF-α (**F**) while decreasing IL-6 (**B**) and IL-12 (**D**), but having no significant effects (*p* > 0.05) on expression of IL-8 (**C**) by macrophages. Levels of expression are presented as median fluorescent intensity (MFI) as determined by flow cytometry. Experiments were carried out in duplicate and repeated 30 times using 30 individual preparations of both monocyte-derived macrophages and DBMSCs. * *p* < 0.05. Bars represent standard errors.

**Figure 6 cells-08-00173-f006:**
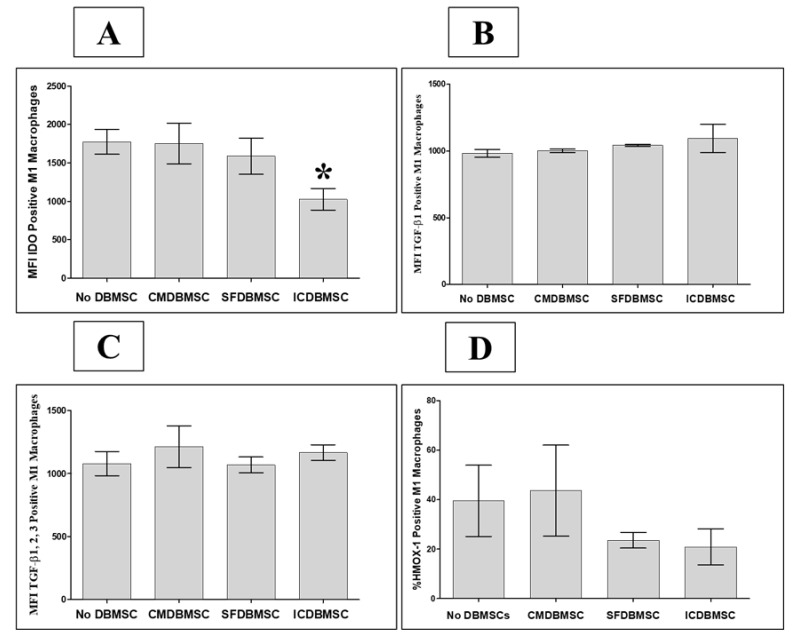
Effects of human DBMSCs on expression of the intracellular molecules IDO, TGFβ1, TGFβ1, 2, 3, and HMOX-1 by human monocytes differentiated into macrophages by GM-CSF, as analyzed by flow cytometry. After six days in culture, compared to untreated macrophages, CMDBMSCs, and SFDBMSCs had no significant effects (*p* > 0.05) on expression of IDO (**A**), TGFβ1 (**B**), TGFβ1, 2, 3 (**C**), and HMOX-1 (**D**) by macrophages. In addition, after six days in culture, compared to untreated macrophages, ICDBMSCs significantly decreased expression by macrophages of IDO (**A**) while having no significant effects (*p* > 0.05) on expression of TGFβ1 (**B**), TGFβ1, 2, 3 (**C**), and HMOX-1 (**D**). Levels of expression are presented as median fluorescent intensity (MFI) or median percentage of HMOX-1 positive cells as determined by flow cytometry. Experiments were carried out in duplicate and repeated 30 times using 30 individual preparations of both monocyte-derived macrophages and DBMSCs. * *p* < 0.05. Bars represent standard errors.

**Figure 7 cells-08-00173-f007:**
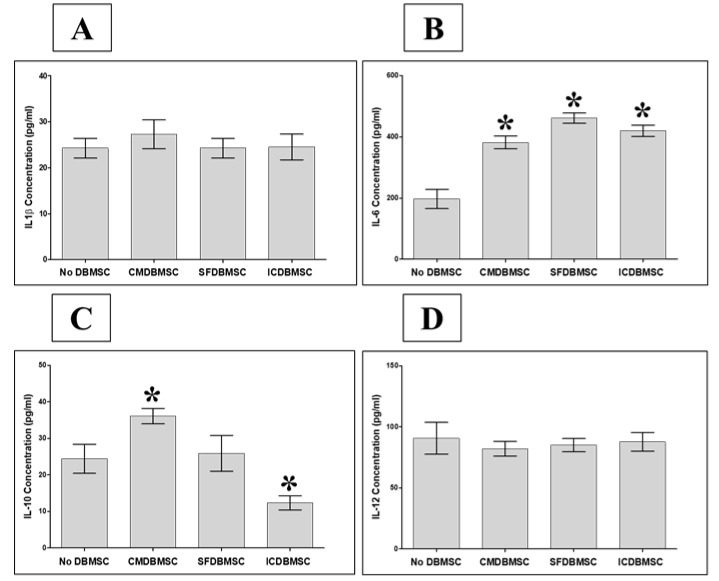
Effects of DBMSCs on GM-CSF–stimulated human monocyte-to-macrophage differentiation, as analyzed by the secretion of IL-1β, IL-6, IL-10, and IL-12 using sandwich ELISA. CMDBMSCs had no significant effect (*p* > 0.05) on the secretion profiles of IL-1β (**A**), significantly increased (* *p* < 0.05) the secretion profiles of IL-6 (**B**) and IL-10 (**C**), and had no significant effect (*p* > 0.05) on the secretion profiles of IL-12 (**D**) of macrophages. In addition, SFDBMSCs had no significant effect (*p* > 0.05) on the secretion profiles of IL-1β (**A**), significantly increased (* *p* < 0.05) the secretion profiles of IL-6 (**B**) and had no significant effect (*p* > 0.05) on the secretion profiles of IL-10 (**C**) and IL-12 (**D**). Moreover, ICDBMSCs had no significant effect (*p* > 0.05) on the secretion profiles of IL-1β (**A**), significantly increased (* *p* < 0.05) the secretion profiles of IL-6 (**B**), significantly decreased (* *p* < 0.05) the secretion profiles of IL-10 (**C**), and had no significant effect (*p* > 0.05) on the secretion profiles of IL-12 (**D**). Experiments were carried out in duplicate and repeated 10 times using 10 individual preparations of both monocyte-derived macrophages and DBMSCs. Bars represent standard errors.

**Figure 8 cells-08-00173-f008:**
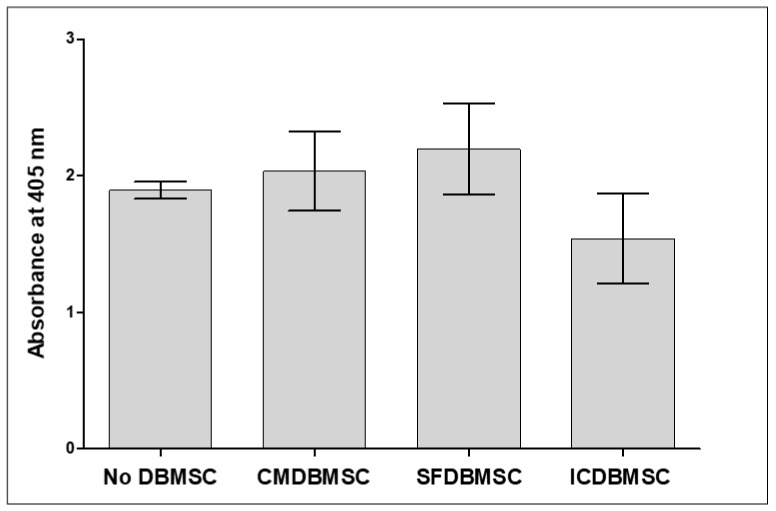
Functional assay of the phagocytic activity of monocytes differentiated into macrophages by GM-CSF in the presence of human DBMSCs. Phagocytosed Zymosan particles were measured by reading the optical density at 405 nm. Phagocytosis of Zymosan particles by macrophages was not significantly affected by DBMSC treatments compared to macrophages cultured alone. Experiments were carried out in duplicate and repeated 10 times using 10 individual preparations of both monocyte-derived macrophages and DBMSCs. Bars represent standard errors.

**Figure 9 cells-08-00173-f009:**
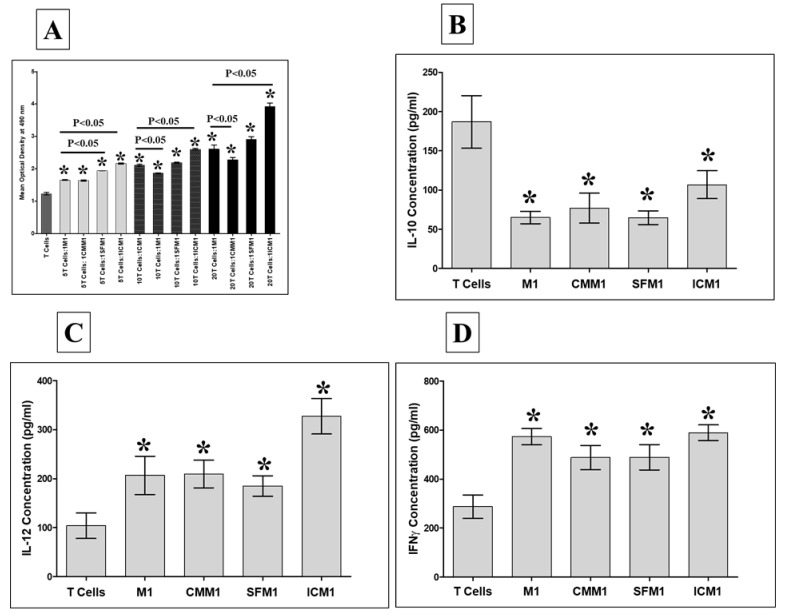
T cell proliferation by M1-like macrophages (M1) generated in the presence of DBMSCs. After culturing M1 alone or with CMDBMSC (CMM1), SFDBMSC (SFM1), and ICDBMSC (ICM1), M1 were harvested and added to allogeneic CD4^+^ T cells at 5:1, 10:1, and 20:1 T cells:M1 (M1, CMM1, SFM1, and ICM1) ratios. M1 pre-treated with DBMSCs (CMM1, SFM1, and ICM1) or without (M1) significantly increased T cell proliferation at all indicated ratios. Similarly, ICM1 significantly increased T cell proliferation at all indicated ratios compared to that with M1, while SFM1 significantly increased T cell proliferation at the 5:1 T cells: SFM1 ratio compared to that with M1. By contrast, CMM1 significantly decreased T cell proliferation at the 10:1 and 20: 1 T cells: CMM1 ratios compared to that with M1 (**A**). In addition, the secretion of IL-10 by T cells stimulated with M1, CMM1, SFM1, and ICM1 was significantly reduced (**B**), while the secretion of IL-12 and IFN-γ was significantly increased, *p* < 0.05 (**C** and **D**). Experiments were carried out in triplicate and repeated 10 times using 10 individual preparations of T cells and macrophages harvested from 10 individual experiments of macrophages cultured with DBMSCs (CMDBMSC, SFDBMSC, and ICDBMSC). * *p*< 0.05, Bars represent standard error.

**Table 1 cells-08-00173-t001:** Markers used in this study.

Macrophage Markers	Costimulatory and HLA Molecules	Inflammatory Molecules	Anti-inflammatory Molecules
CD14	CD40	IL-1β	IDO
CD11b	CD80	IL-6	TGFβ1
CD36	CD86	IL-8	TGFβ1, 2, 3
CD163	CD273	IL-12	HMOX-1
CD204	CD274	IFN-γ	
CD206	HLA-DR	TNF-α	
B7-H4			

**Table 2 cells-08-00173-t002:** DBMSC (CMDBMSC, SFDBMSC, and ICDBMSC) effects on macrophage expression of different functional markers, costimulatory molecules, co-inhibitory molecules, antigen presenting molecule, inflammatory cytokines, and anti-inflammatory markers as compared to untreated macrophages by flow cytometry recorded by mean of fluorescent intensity (MFI). Increased (↑, *p* < 0.05), decreased (↓, *p* < 0.05)), and no change (*p* > 0.05).

Markers	Types	CMDBMSC	SFDBMSC	ICDBMSC		Markers	Types	CMDBMSC	SFDBMSC	ICDBMSC
CD14	**Functional Markers**	↑	↑	No Change		IDO	**Anti-inflammatory Markers**	No Change	No Change	↓
CD163		↑	↑	↓		TGFβ1		No Change	No Change	No Change
CD204		No Change	No Change	↓		TGFβ1, 2, 3		No Change	No Change	No Change
CD206		No Change	↑	↓		HMOX-1		No Change	No Change	No Change
CD36		No Change	No Change	↓						
B7H4		No Change	No Change	No Change						
CD80	**Costimulatory Molecules**	No Change	No Change	No Change						
CD86		↓	↓	↓						
CD273	**Co-Inhibitory Molecules**	No Change	No Change	↓						
CD274		No Change	No Change	↓						
HLA-DR	**Antigen Presenting Molecule**	No Change	No Change	No Change						
IL-1β	**Inflammatory Cytokines**	↓	↓	↑						
IL-6		No Change	No Change	↓						
IL-8		No Change	No Change	No Change						
IL-12		No Change	↓	↓						
IFN-γ		No Change	↑	↑						
TNF-α		No Change	No Change	↑						

**Table 3 cells-08-00173-t003:** DBMSC (CMDBMSC, SFDBMSC, and ICDBMSC) effects on macrophage secretion of inflammatory and anti-inflammatory cytokines as compared to untreated macrophages by ELISA. Increased (↑) and decreased (↓).

Markers	Type	CMDBMSC	SFDBMSC	ICDBMSC
IL-1β	**Inflammatory Cytokines**	No Change	No Change	No Change
IL-6		↑	↑	↑
IL-10		↑	No Change	↓
IL-12	**Anti-inflammatory Cytokine**	No Change	No Change	No Change

**Table 4 cells-08-00173-t004:** DBMSC (CMDBMSC, SFDBMSC, and ICDBMSC) effects on macrophage stimulatory effects on T cell proliferation and secretion of inflammatory and anti-inflammatory cytokines as compared to untreated T cells measured by ELISA. Increased (↑) and decreased (↓).

Markers	Type	CM1	CMM1	SFM1	ICM1
IL-12	Inflammatory Cytokine	↑	↑	↑	↑
IFN-γ					
IL-10	Anti-inflammatory Cytokine	↓	↓	↓	↓

## Data Availability

All data generated or analyzed during this study are included in this published article.

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
