# Peer review of "Decidua Basalis Mesenchymal Stem Cells Favor Inflammatory M1 Macrophage Differentiation In Vitro"

_cells, 2019, doi:10.3390/cells8020173_

Round 1

Reviewer 1 Report

The results shown are rather interesting and deserve publication.

The experimental part is sound. No further data are required

Author Response

Thank you very much for reviewing our manuscript. We strongly believe that our manuscript is well written and it does not require further English editing.

Reviewer 2 Report

General comments:

Nice work showing crosstalk between MSC and macrophages and its relevance to T cell viability. Have a few suggestions to improve its content. Also would recommend to look at and include in the discussion the Waterman et al paper: “A new mesenchymal stem cell (MSC) paradigm: polarization into a pro-inflammatory MSC1 or an Immunosuppressive MSC2 phenotype. Waterman RS, Tomchuck SL, Henkle SL, Betancourt AM. PLoS One. 2010; 5(4):e10088. doi: 10.1371/journal.pone.0010088.

Specific comments:

Line 16: is there a professional email that could be used instead of a personal Hotmail?

Line 41: may need to use “favor” instead of “favour”

Line 56: can you specify what type of interaction is “DBMSC with monocytes”

Line 57: is it “did modify” instead of “did not modify”

Line 61: mention what type of T cell? In “T cell proliferation”

Line 62: may need to use: “enhance” instead of “favor” in: “We showed that DBMSC favor”

Line 64: may need to use: “enhance” or “enhancement” instead of “induction”

Line 65: may need to remove: “However, future study is essential to confirm this”

Line 111: may need to remove the word: “(angiogenesis)”

Line 116: add “whether” after the word “examined”

Line 119: may need to change “favored” to “enhanced”

Line 164: another way to polarize macrophages into M1 phenotype is by adding LPS and IFNγ. Could authors comment how different is their M1 polarization from others in which primary monocytes are differentiated with MCSF for 6 days and treated with LPS and IFNγ.to generate primary M1 macrophages.

Line 199: is it “1:20” instead of “20:1”?

Figure 1: comparing panels “C” or “D” to either “A” or “B”, there is higher number of “spindle-like” macrophages. Could you quantify that and comment on it on the discussion?

Line 251: what does this ratio refer to? Is it a cell ratio? If so, then this may need to be mentioned

Line 272: what is the rationale of measuring these markers by flow cytometry?

Line 283: why is the effect of CMDBMSC different from SFMSC?

Line 308: what are the levels of GMSF in the collected conditioned media from DBMSC?

Line 332: can you explain the difference between IC and DC coculture on CD206 levels?

Line 335:may need to change “B7H7” into “B7-H4”

Lines 345 and 346: could you link changes in HLA-Dr, CD80 and CD247 to changes in CD206?

Line 380 (table 2): what was the cut-off (i.e. fold change) for a decrease or increase and P values?

Table 2: what does the HMOX-1 marker refer to? May need to write something under “Types”

Line 388: add “)” after letter “(F.

Line 415: what is relevance of decreased IDO levels in macrophages?

Line 418: may need to change “HO-1” into “HMOX-1”

Line 446: would be good to have a positive control for this phagocytic assay (at least show that in a supplemental figure how it works)

Line 475: remove “s” from “functions”

Line 478: and throughout rest of the text: may need to rewrite “CM1” as just “M1” that would avoid confusion since CM could also mean conditioned media.

Line 479: did you try a 1:1 ratio of T cell : M1-like macrophages

Line 498: again, may need to label “CM1” and just “M1” macrophages here and any where else in the text, figure legends, and figures. This may help avoid confusion since CM could also mean conditioned media

Lines 500 and 501: could you explain why SFM1 increased T cell proliferation relative to CM1

Author Response

General comments:

Nice work showing crosstalk between MSC and macrophages and its relevance to T cell viability. Have a few suggestions to improve its content. Also would recommend to look at and include in the discussion the Waterman et al paper: “A new mesenchymal stem cell (MSC) paradigm: polarization into a pro-inflammatory MSC1 or an Immunosuppressive MSC2 phenotype.Waterman RS, Tomchuck SL, Henkle SL, Betancourt AM. PLoS One. 2010; 5(4):e10088. doi: 10.1371/journal.pone.0010088.

Answer

Inserted “This phenomenon of monocyte polarization into immunostimulatory or immunoinhibitory macrophages was also reported for MSCs. It was shown that MSCs can be polarized into immunostimulatory or immunosuppressive cells similar to monocytes (1)”.

Specific comment 1:

Line 16: is there a professional email that could be used instead of a personal Hotmail?

 Answer

Unfortunately, Dr Mohammed. F. El- Muzaini retired, and his personal Hotmail is only available.

Specific comment 2:

Line 41: may need to use “favor” instead of “favour”

Answer

Corrected (favor).

Specific comment 3:

Line 56: can you specify what type of interaction is “DBMSC with monocytes”

Answer

Changed to “The culture of DBMSCs with monocytes did not inhibit monocytes differentiation into M1-like inflammatory macrophages.” to make it clearer.

 Specific comment 4:

Line 57: is it “did modify” instead of “did not modify”

Answer

Corrected to “did not inhibit

 Specific comment 5:

Line 61: mention what type of T cell? In “T cell proliferation”

Answer

Changed to “CD4+ T cell”.

Specific comment 6:

Line 62: may need to use: “enhance” instead of “favor” in: “We showed that DBMSC favor”

Answer

Changed to “enhanced”.

 Specific comment 7:

Line 64: may need to use: “enhance” or “enhancement” instead of “induction”

Answer

Changed to “enhancement”.

 Specific comment 8:

Line 65: may need to remove: “However, future study is essential to confirm this”

Answer

Removed as suggested.

Specific comment 9:

Line 111: may need to remove the word: “(angiogenesis)”

Answer

Removed as suggested.

 Specific comment 10:

Line 116: add “whether” after the word “examined”

Answer

Added as suggested “whether”.

Specific comment 11:

Line 119: may need to change “favored” to “enhanced”

Answer

Changed to “enhanced” as suggested.

 Specific comment 12:

Line 164: another way to polarize macrophages into M1 phenotype is by adding LPS and IFNγ. Could authors comment how different is their M1 polarization from others in which primary monocytes are differentiated with MCSF for 6 days and treated with LPS and IFNγ.to generate primary M1 macrophages.

Answer

We can use LPS and IFNγ to polarize macrophages into M1 phenotype. To achieve this monocytes will be first incubated with PMA for 24 hours to induce M0 phenotype. M0 macrophages will then be cultured alone in fresh RPMI-164 0 medium. After 24 hour, LPS and IFNγ will be added to M0 macrophages to induce M1 phenotype.

 In contrast, to polarize monocytes into M1 phenotype using GMCSF (as we did), it does not require the use of PMA to induce M0 phenotype. Instead, monocytes are continuously incubated with GMCSF for 6 days. On day 6, M1 macrophages will appear.

 Specific comment 13:

Line 199: is it “1:20” instead of “20:1”?

Figure 1: comparing panels “C” or “D” to either “A” or “B”, there is higher number of “spindle-like” macrophages. Could you quantify that and comment on it on the discussion?

 Answer

Corrected to “1:20”.

In Panels C and D, the total number of spindle like shaped cells is less than <0.3% and therefore it is neglected.

Specific comment 14:

Line 251: what does this ratio refer to? Is it a cell ratio? If so, then this may need to be mentioned

Answer

It means cell ratio, and it is already mentioned by saying “monocyte: DBMSC ratio”.

However, we added cell ratios”.

Specific comment 15:

Line 272: what is the rationale of measuring these markers by flow cytometry?

Answer

Flow cytometry is a highly quantitative method that can answer biological questions in more details and faster than other protein quantitative methods.  

Specific comment 16:

Line 283: why is the effect of CMDBMSC different from SFMSC?

Answer

Using conditioned medium from unstimulated DBMSCs (CMDBMSC) means that DBMSCs produced factors under no stimulatory effect while using the paracrine culture system (SFDBMSC) means that DBMSC and macrophages together contributed to the production of factors. Therefore, factors produced using the CMDBMSC are different in their composition and concentrations than the factors produced using the SFDBMSC culture system. Consequently, it is expected that the effect of CMDBMSC will be different than SFDBMSC on macrophages.

Specific comment 17:

Line 308: what are the levels of GMSF in the collected conditioned media from DBMSC?

Answer

 We assume that it would be around 26.50 ± 6.43 pg/mL as we previously published (2).

 Specific comment 18:

Line 332: can you explain the difference between IC and DC coculture on CD206 levels?

Answer

If I understood the question correctly “The difference between the effects of IC and SF on CD206 level”. It will be attributed to the culture system.

Using the IC (Intercellular culture system), there is physical cellular communication via the pores of the membrane while using the SF (Soluble factor) culture system, the effect is based on soluble factors not cellular contact. Therefore, this may contribute to the difference in CD206 level between the two culture systems.

 Specific comment 19:

Line 335: may need to change “B7H7” into “B7-H4”

Answer

Corrected “B7-H7”.

 Specific comment 20:

Lines 345 and 346: could you link changes in HLA-Dr, CD80 and CD247 to changes in CD206?

Answer

Added “co-inhibitory molecules (CD273 and CD274)”.

Added “HLA-DR (antigen presenting molecule) and CD80 (co-stimulatory molecule)”.

Added “These data indicate that ICDBMSC maintained the antigen presentation activity of macrophages and with immunostimulatory effect by maintaining the expression of co-stimulatory molecule (CD80) and reducing the expression of the co-inhibitory molecules (CD273 and CD274)”.

 Specific comment 21:

Line 380 (table 2): what was the cut-off (i.e. fold change) for a decrease or increase and P values?

Answer

It is already presented in Figures 5, 6 and their legends. Added “Increased (é, P<0.05), decreased (ê, P<0.05)), and no change (P>0.05)”.

Specific comment 22:

Table 2: what does the HMOX-1 marker refer to? May need to write something under “Types”

Answer

Corrected “Anti-inflammatory Markers

Specific comment 23:

Line 388: add “)” after letter “(F.

Answer

Corrected “(F)”.

Specific comment 24:

Line 415: what is relevance of decreased IDO levels in macrophages?

Answer

It is already discussed in the discussion section “In addition, ICDBMSCs decreased macrophage expression of IDO (Figure 6A). IDO is an immunosuppressive enzyme that stimulates the catabolism of tryptophan, inhibiting the immune responses by repressing T cell function”.

Specific comment 25:

Line 418: may need to change “HO-1” into “HMOX-1”

Answer

Corrected “HMOX-1”.

Specific comment 26:

Line 446: would be good to have a positive control for this phagocytic assay (at least show that in a supplemental figure how it works)

Answer

The positive control is represented by “No DBMSC” as shown in Figure 8 and discussed in the legend “Phagocytosis of Zymosan particles by macrophages”. This assay relies on the phagocytosis of Zymosan particles so it is serving as a positive control.

Specific comment 27:

Line 475: remove “s” from “functions”

Answer

Corrected “function

 Specific comment 28:

Line 478: and throughout rest of the text: may need to rewrite “CM1” as just “M1” that would avoid confusion since CM could also mean conditioned media.

Answer

Corrected as suggested

 Specific comment 29:

Line 479: did you try a 1:1 ratio of T cell : M1-like macrophages

Answer

No.

Specific comment 30:

Line 498: again, may need to label “CM1” and just “M1” macrophages here and any where else in the text, figure legends, and figures. This may help avoid confusion since CM could also mean conditioned media

Answer

Corrected as suggested

Specific comment 31:

Lines 500 and 501: could you explain why SFM1 increased T cell proliferation relative to CM1

Answer

As explained above and already discussed “These data show the experimental conditions (mechanism of DBMSC communication with monocytes) modulate the phenotypic characteristics (inflammatory or anti-inflammatory phenotypes) of differentiated macrophages”, and “Additionally, we showed that paracrine communication between DBMSCs (SFDBMSC) and monocyte-derived macrophages increased the expression of IFN-γ (Figure 5E) and secretion of IL-6 (Figure 7B) while decreasing the expression of IL-1β and IL-12 (Figure 5A and D) by macrophages. Likewise, soluble mediators secreted by unstimulated DBMSCs (CMDBMSC) decreased the expression of IL-1β (Figure 5A) and increased the secretion of IL-6 and IL-10 by macrophages (Figure 7B and C). Collectively, our data indicate that DBMSCs may exert opposing effects on the phenotype of macrophages”.

Reviewer 3 Report

Given MSC generally are immunomodulators and reduce T cell proliferation, how do you explain that DBMSC induce proliferation of CD4+ T cells?

In figure 1G-L it shows that increasing amounts of DBMSC inhibits macrophage differentiation as you see the presence of monocytes. Doesn't this go against the idea that they promote M1 macrophage differentiation?

Author Response

Comment 1:

Given MSC generally are immunomodulators and reduce T cell proliferation, how do you explain that DBMSC induce proliferation of CD4+ T cells?

 Answer 1:

DBMSCs enhanced M1 macrophage differentiation, immune cells with immunostimulatory property and this subsequently induced the proliferation of CD4+ T cells as mentioned in the discussion section “Activated M1-like macrophages usually phagocytose microbes, remove tumor cells, present antigen to T cells to trigger an immune response, and secrete high levels of inflammatory cytokines” and “DBMSC treatments induced stimulatory effects of M1-like macrophages on the proliferation (Figure 9A) and production of high levels of inflammatory cytokines, including IL-12 and IFN-γ, and low levels of IL-10 (Figure 9B-D) by CD4+ T cells.

Comment 2:

In figure 1G-L it shows that increasing amounts of DBMSC inhibits macrophage differentiation as you see the presence of monocytes. Doesn't this go against the idea that they promote M1 macrophage differentiation?

Answer

No. It means that DBMSC enhancement of M1 macrophages is decreasing with increasing concentration.